# Toward Learning Generalized Cross-Problem Solving Strategies for Combinatorial Optimization

## Abstract

Combinatorial optimization (CO) problems are fundamental across various domains, with many sharing similarities in optimization objectives, decision variables, and constraints. Many traditional algorithms perform well on related problems using similar solution strategies, highlighting the commonality in solving different problems. However, most machine learning approaches treat each CO problem in isolation, failing to capitalize on the underlying relationships between problems. In this paper, we investigate the potential to learn generalized solving strategies that capture the shared structure among different CO problems, enabling easier adaptation to related tasks. To this end, we propose to first divide the model architecture into three components: a header, an encoder, and a decoder; where The header and decoder address problem-specific inputs and outputs, while the encoder is designed to learn shared strategies that generalize across different problems. To ensure this, we enforce alignment in the optimization directions of the encoder across problems, maintaining consistency in both gradient directions and magnitudes to harmonize optimization processes. This is achieved by introducing the additional problem-specific rotation matrices and loss weights to steer the gradients, which are updated via a gradient consistency loss. Extensive experiments on six CO problems demonstrate that our method enhances the model's ability to capture shared solving strategies across problems. We show that the learned encoder on several problems can directly perform comparably on new problems to models trained from scratch, highlighting its potential to support developing the foundational model for combinatorial optimization. Source code will be made publicly available.

## 1 Introduction

Combinatorial optimization (CO) problems, which involve optimizing discrete variables under specific objectives, are a fundamental class of problems in computer science, operations research, and beyond, serving widespread applications in practical scenarios Hong et al. (2010); Mironov & Zhang (2006); Ganzinger et al. (2004). However, due to their inherent computational complexity, e.g. NP-hardness, solving efficiency poses significant challenges and requires exhaustive human efforts to design solving heuristics. Traditionally, this has necessitated the development of problem-specific heuristics, requiring exhaustive human effort and domain expertise. Recently, advances in machine learning (ML) have demonstrated the potential to automatically learn solving heuristics from data, improving both solution quality and speed when the problem instances fall within certain distributions (Bengio et al., 2021; Kool et al., 2018; Joshi et al., 2019; Kwon et al., 2020; Sun & Yang, 2023; Li et al., 2023). In addition, learning can help quickly uncover new heuristics for new problems or new instance distributions where experts are not there.

Despite their complexity, many CO problems share common structures, exhibiting similarities in optimization objectives, decision variables, or constraints (Kool et al., 2018; Kwon et al., 2020). This shared nature extends even further in the realm of NP-complete (NPC) problems, where reductions between problems reveal a deeper computational equivalence (Cook, 2023; Karp, 2010). The presence of these commonalities suggests that techniques designed for one CO problem may have broader relevance, offering the potential for more generalizable and adaptable solutions. Historically, classical heuristics (Papadimitriou & Steiglitz, 2013; Dorigo et al., 2006) have demonstrated this versatility,

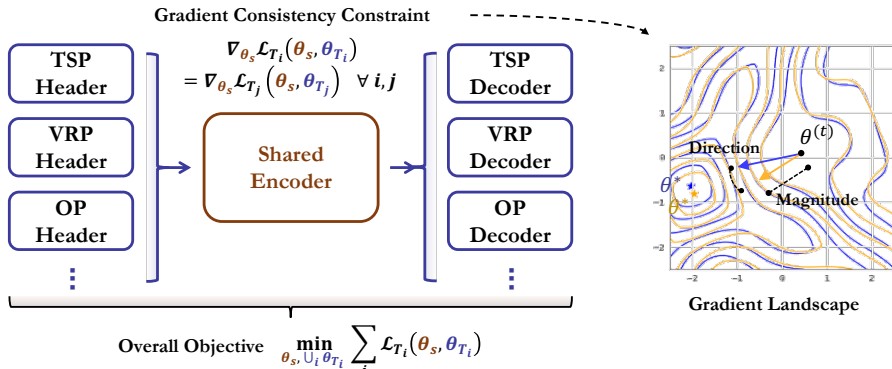

Figure 1: The GCNCO framework utilizes light-parametered headers and decoders to handle problem-specific inputs and outputs, respectively, and enforces a gradient consistency constraint on the heavy-parametered shared encoder to extract generalized strategies. The gradient consistency constraint involves homogenizing the gradients of different problems regarding direction and magnitude.

proving effective across various related problems with minimal adaptation. This observation raises the possibility that machine learning models, when trained on diverse CO problems, could similarly capture shared knowledge across tasks, learning a generalized problem-solving strategy. Such an approach could enable these models to perform well on new, unseen problems with minimal task-specific fine-tuning.

To this end, we introduce gradient-consistent neural combinatorial optimization (GCNCO) to enforce the consistency of the optimization trajectories across tasks and promote learning generalizable strategies. Gradients are the driving force behind model optimization, guiding it through the landscape of a given task (Ruder, 2016; LeCun et al., 2015). If a models optimization trajectory, reflected in the gradient directions and magnitudes, remains consistent across tasks, it suggests that the model is learning a shared, underlying solution strategy that is applicable across different CO problems. Gradient consistency ensures that when optimizing for one problem, the models learned parameters improve performance not just for that specific task but also for others. This alignment of gradients across tasks creates a unified approach to solving CO problems, preventing the model from overfitting to one particular task and instead promoting a general-purpose strategy.

Specifically, we first introduce a header-encoder-decoder framework to allow the model to process different problems in a unified manner, where the header and decoder handle problem-specific inputs and outputs, and the encoder is designed to learn a common strategy that generalizes across multiple CO problems. Then, we propose to align the optimization directions of the encoder across different problems, ensuring consistency in both gradient directions and magnitudes during training. The encoder gradients of different tasks are steered by newly introduced task-specific feature rotation matrices and loss weights. The feature rotation matrices rotate the optimization landscape and thus adjust the gradient direction, while the loss weights modulate the gradient magnitude. The whole framework follows a bi-level optimization process where the original models are optimized through the solving objective of different problems, and the additional rotation matrices and loss weights are optimized through a gradient-consistent loss reflecting the gradient homogenization. This approach ensures that the encoder learns strategies that are not only effective for individual tasks but also applicable across a range of CO problems.

Our experiments cover six combinatorial optimization problems: Travelling Salesman Problem (TSP), Vehicle Routing Problem (VRP), Split Delivery VRP (SDVRP), Orienteering Problem (OP), Prize Collecting TSP (PCTSP), and Stochastic PCTSP (SPCTSP). First, we analyze the relationships between tasks by studying the gradient alignment across problems. We then demonstrate the effectiveness of our method in learning a general solving strategy by evaluating its performance on both training tasks and new, unseen problems under zero-shot evaluation and finetuning settings. Remarkably, we show that the learned encoder, when applied to new problems, achieves results comparable to models trained from scratch, highlighting the potential of our approach to serve as a foundational model for combinatorial optimization.

## 2 RELATED WORK

**Neural Combinatorial Optimization.** Recent advancements in machine learning for combinatorial optimization (CO) have introduced innovative solvers that can be broadly categorized into constructive and improvement-based approaches. Constructive methods, including autoregressive techniques Khalil et al. (2017); Kool et al. (2018); Kwon et al. (2020); Hottung et al. (2021); Kim et al. (2022), sequentially determine decision variables to build complete solutions, while non-autoregressive strategies Joshi et al. (2019); Fu et al. (2021); Geisler et al. (2022); Qiu et al. (2022); Sun & Yang (2023) generate soft-constrained solutions in a single pass, subsequently refined through post-processing for feasibility. In contrast, improvement-based solvers d O Costa et al. (2020); Wu et al. (2021); Chen & Tian (2019); Li et al. (2021); Hou et al. (2023) focus on iteratively enhancing existing solutions using local search operators to optimize the objective. The promise of generative modeling in CO lies in its robust representational abilities and effective distribution estimation, framing problem-solving as a conditional generation task to learn solution distributions tailored to specific instances Hottung et al. (2021); Cheng et al. (2022); Sun & Yang (2023); Du et al. (2023); Zhang et al. (2023); Li et al. (2023).

**Multi-task learning**. Multi-Task Learning (MTL) seeks to boost the effectiveness of multiple tasks by training a unified model that captures the shared insights across these tasks. A plethora of studies have tackled MTL from different angles, such as striking a balance in the loss functions of diverse tasks (Mao et al., 2021; Dai et al., 2023), developing mechanisms for module sharing (Javaloy & Valera, 2021), and leveraging meta-learning (Wang et al., 2021). In an effort to enhance the efficiency of MTL and reduce the detrimental effects of negative transfer (Jiang et al., 2024), some research has turned to task-grouping strategies (Fifty et al., 2021), aiming to discern task relationships and facilitate learning within these groups to minimize negative transfer between conflicting tasks. MTL has found widespread application in many fields (Allenspach et al., 2024; Zhang et al., 2024). Nonetheless, there is a scarcity of research on applying MTL to solve COPs.

**Multi-task Learning for CO.** The unified model capable of simultaneously addressing multiple CO problems is still in its early stages. The ability of neural solvers to generalize across tasks has garnered increasing interest. Ibarz et al. (2022) introduce a generalist neural algorithmic learner a single graph neural network processor designed to execute a broad range of algorithms for classic polynomial problems. For more computationally complex CO problems, Wang & Yu (2023) propose a multi-armed bandit approach to solve multiple CO problems by alternating optimizing different problems. For specialized problems like Vehicle Routing Problems (VRPs), Lin et al. (2024) suggest training a shared backbone model for VRPs, which can then be fine-tuned efficiently for different VRP variants using linear projections. Additionally, Zhou et al. (2024); Liu et al. (2024); Berto et al. (2024) apply attribute composition to achieve (zero-shot) generalization across various VRP variants. While these approaches treat a set of problem variants as extensions of a single-task model with varying attributes, this paper explores learning generalized strategies across different problems, aiming to prune the single-task model for the generalized part rather than expanding it.

## 3 PRELIMINARIES

Adopting the conventions established in (Karalias & Loukas, 2020; Wang et al., 2022), we define $\mathcal{G}$ as the collection of CO problem instances represented by graphs $G(V, E) \in \mathcal{G}$, where $V$ and $E$ denote the nodes and edges respectively. CO problems can be broadly classified into two types based on the solution composition: edge-decision problems that involve determining the selection of edges and node-decision problems that determine nodes. Let $\mathbf{x} \in \{0, 1\}^N$ denote the optimization variable, where each entry indicates whether it is included in the solution. For edge-decision problems, $N = n^2$ and $\mathbf{x}_{i,j}$ indicates whether $E_{i,j}$ is included in $\mathbf{x}$. For node-decision problems, $N = n$ and $\mathbf{x}_i$ indicates whether $V_i$ is included in $\mathbf{x}$. The feasible set $\Omega$ consists of $\mathbf{x}$ satisfying specific constraints as feasible solutions. A CO problem on $G$ aims to find a feasible $\mathbf{x}$ that minimizes the given objective function $l(\cdot; G) : \{0, 1\}^N \to \mathbb{R}_{\geq 0}$:

$$\min_{\mathbf{x} \in \{0,1\}^N} l(\mathbf{x}; G) \quad \text{s.t.} \quad \mathbf{x} \in \Omega \tag{1}$$

Yet different problems maintain differences in loss functions and constraints. In this paper, we consider the following problems, including the Traveling Salesman Problem (TSP), which is an

edge-decision problem where the objective is to find the shortest tour that visits every node exactly once and returns to the starting point. The Vehicle Routing Problem (VRP), more specifically the Capacitated VRP (CVRP), requires determining routes that start and end at a depot while ensuring the total demand on each route does not exceed vehicle capacity. The Split Delivery VRP (SDVRP) extends this by allowing deliveries to be split across multiple routes. The Orienteering Problem (OP) is a variant of TSP that aims to maximize the total prize collected from visited nodes under a distance constraint without needing to visit all nodes. For the Prize Collecting TSP (PCTSP), each node not only has a prize but also a penalty for being unvisited, and the goal is to collect a minimum total prize while minimizing both the tour length plus the penalties of unvisited nodes. The Stochastic PCTSP (SPCTSP) introduces uncertainty, where the expected prize is known upfront, while the actual prize is only revealed upon visitation. Each of these problems presents unique challenges and variations in the loss functions and constraints.

# 4 GRADIENT-CONSISTENT NEURAL COMBINATORIAL OPTIMIZATION

## 4.1 OPTIMZIATION FOR GENERALIZED STRATEGIES

Consider a set of tasks $\{T_i\}_{i=1}^k$, where the goal is to learn a neural network solver $S_\theta$ that can map $X_i$ to $Y_i$, i.e., $S_\theta : X_i \rightarrow Y_i$. The primary aim is to learn a generalizable solution strategy that captures the shared information across multiple tasks while also maintaining task-specific capabilities. As a result, this model possesses the ability to solve a variety of problems in a generalized manner, with the learned strategies being effective across multiple tasks, which may potentially be effective for new, unseen tasks. Furthermore, this model can serve as a pre-trained model, where its performance on a specific task can be optimized through tuning.

To achieve this, we structure the model with a shared network $\theta_s$ responsible for learning generalized strategies, supplemented by task-specific modules $\theta_{T_i}$ designed to handle the unique input and output form of each task. The complete model is represented as $\{\theta_s, \theta_{T_i}, \ldots, \theta_{T_k}\}$, where for a particular task $T_i$, the parameters involved are $\{\theta_s, \theta_{T_i}\}$. The shared network $\theta_s$ captures knowledge common across tasks, while the task-specific modules $\theta_{T_i}$ ensure sufficient specialization for each problem.

The optimization objective for learning these generalized strategies is to ensure that, with the support of the task-specific modules $\theta_{T_i}$, the relationship $P(X_i \rightarrow Y_i)$ remains consistent and stable across tasks, reflecting a potential unified problem-solving strategy. This can be expressed as:

$$P(X_i \rightarrow Y_i | \theta_{T_i}) = P(X_j \rightarrow Y_j | \theta_{T_j}), \forall i, j \tag{2}$$

Then, there exists a neural model $\theta_s$ corresponding to this mapping relation, and this consistency in problem-solving strategies across tasks ensures that updates to $\theta_s$ lead to simultaneous improvements across the tasks. In other words, the shared parameters $\theta_s$ should contribute to all tasks in a stable and coherent manner. To enforce this, we introduce a gradient consistency constraint during the optimization process. This constraint ensures that the gradient directions and magnitudes for $\theta_s$ across different tasks remain aligned, thereby guiding the model toward a generalized solution. Specifically, the optimization objective becomes:

$$\min_{\theta_s, \bigcup_i \theta_{T_i}} \sum_i \mathcal{L}_{T_i}(\theta_s, \theta_{T_i})$$
$$s.t. \quad \nabla_{\theta_s} \mathcal{L}_{T_i}(\theta_s, \theta_{T_i}) = \nabla_{\theta_s} \mathcal{L}_{T_j}(\theta_s, \theta_{T_j}), \forall i, j \tag{3}$$

where $\mathcal{L}_{T_i}$ is the loss for problem $T_i$. By enforcing the gradient alignments, we ensure that the shared network learns strategies that are equally beneficial across tasks, promoting both generalization and robustness in the learned solution strategies.

## 4.2 THE GENERAL MODEL DESIGN

In order for the model to be compatible with as many tasks as possible, it needs to have strong adaptability to constraints, meaning it should effectively handle different problem constraints without requiring additional post-processing to ensure solution feasibility. Therefore, we adopt a sequential decision-making approach, where a point is selected at each step to be added to the existing partial solution, ultimately forming a complete solution optimized through reinforcement learning (RL)

to achieve the objective function. This method ensures that constraints are satisfied at each step, thus addressing the differences in constraints across multiple problems in a unified manner. On the other hand, using the original objective function for optimization explicitly provides the model with information about the objective function, allowing it to better capture the differences in objective functions between problems.

Without loss of generality, taking TSP as an example, the solution sequence $\pi = (\pi_1, \pi_2, \dots, \pi_n)$ is treated as a permutation of nodes, depending on the problem type. The model defines a stochastic policy $p_\theta(\pi|s)$ for selecting the sequence solution $\pi$ given a problem instance $G$:

$$p_\theta(\pi|G) = \prod_{t=1}^{n} p_\theta(\pi_t|G, \pi_{1:t-1}), \tag{4}$$

where $\pi_t$ represents the decision at step $t$, conditioned on the current partial solution $\pi_{1:t-1}$ and problem instance $s$. The policy $p_\theta$ is parameterized by $\theta$, typically learned through a graph neural network or attention mechanism. The probability distribution $p_\theta(\pi_t|G, \pi_{1:t-1})$ determines the selection of the next node based on the current graph context and previously selected components. The model can be directly optimized through the expectation of the given problem objective $\mathcal{L}(\theta|G) = \mathbb{E}_{p_\theta(\pi|G)}[L(\pi)]$. We optimize $\mathcal{L}$ by gradient descent, using the REINFORCE () gradient estimator with baseline $b(G)$ following Kool et al. (2018); Kwon et al. (2020):

$$\nabla\mathcal{L}(\theta|G) = \mathbb{E}_{p_\theta(\pi|G)}\left[(L(\pi) - b(G))\nabla \log p_\theta(\pi|G)\right]. \tag{5}$$

To equip the model with the ability to handle various problems, we follow Wang & Yu (2023) to adopt separate headers and decoders for each problem to handle problem-specific inputs and outputs, and the entire model follows a header-encoder-decoder framework, where the headers and decoders consist of a single layer each, while the encoder, with significantly larger parameters, is designed to learn the core solving strategy.

The headers preprocess the raw input of different combinatorial optimization problems, converting information such as coordinates, demands, rewards, and penalties into unified vector representations. The encoding steps of the shared encoder are consistent with the Transformer (Vaswani, 2017), where the input is first mapped to $Q$, $K$, $V$ by passing through the weight matrices $W_q$, $W_k$, $W_v$ of linear layers. Then, multi-head attention is used to capture the correlations between input nodes. The output is combined with the input through residual connections and layer normalization, followed by processing with a feed-forward neural network to extract features for each task. The decoding steps in the Decoder differ from the Transformer decoder, combining multi-head attention and single-head attention. Before selecting actions in the final step, attention scores and embedding vectors are generated based on the current problem state (such as the current node, remaining capacity, etc.) using multi-head attention. These intermediate variables are then processed through single-head attention to compute the probability of an action rather than generating a continuous sequence. This simplifies the action selection process, directly outputting the probability distribution for action selection, making it more suitable for specific optimization tasks.

### 4.3 Gradient Homogenization

Since we expect the encoder to learn a unified strategy that is effective across all tasks, corresponding to the relationship $P(X_i \rightarrow Y_i|\theta_{T_i})$, the optimization gradient landscape of the network should remain consistent across different tasks during optimization. However, since this gradient field is difficult to estimate and cannot be directly used in the loss function computation, we attempt to standardize the gradient vectors at each step along the optimization trajectory. Specifically, we homogenize the gradients of the encoder in a header-encoder-decoder architecture regarding gradient directions and magnitudes.

**Gradient Magnitude Homogenization.** We adopt a hyperparameter-free method that normalizes the gradient magnitudes across tasks, following Normalized Gradient Descent (Cortés, 2006) and Javaloy & Valera (2021). Let the shared representation at the encoder output be denoted by $\mathbf{z}$. Then the feature-level gradients of the task $T_i$ for the $k$-th data point be denoted as $\mathbf{g}_{T_i,k} = \nabla_{\mathbf{z}_k}\mathcal{L}_{T_i}(\mathbf{z}_k)$, and the batch gradient for task $T_i$ is represented as $\mathbf{G}_{T_i}^\top := [\mathbf{g}_{T_i,1}, \mathbf{g}_{T_i,2}, \dots, \mathbf{g}_{T_i,B}]$, where $B$ is the batch size. Then, we can intuitively rescale the gradients of different problems through the normalization:

$$\mathbf{U}_{T_i} = \frac{\mathbf{G}_{T_i}}{\|\mathbf{G}_{T_i}\|}, \forall i \tag{6}$$

This operation balances the gradient magnitudes across different tasks. However, there exists a difference in the magnitude of the gradients relative to before normalization. Therefore, we introduce a parameter $C$ representing the scalar target magnitude for all task gradients to balance the difference between the gradients before and after normalization, and the final gradients become $C\mathbf{U}_{T_i}$.

We define $C$ as a convex combination of task-specific gradient magnitudes at iteration $t$:

$$C := \sum_i \alpha_{T_i} \|\mathbf{G}_{T_i}\|, \tag{7}$$

where the weights $\alpha_{T_i}$ are adaptive and reflect the relative convergence speed of each task. The weights $\alpha_{T_i}$ sum to 1 and are defined as:

$$\alpha_{T_i} = \frac{\|\mathbf{G}_{T_i}\|/\|\mathbf{G}_{T_i}^0\|}{\sum_j \|\mathbf{G}_{T_j}\|/\|\mathbf{G}_{T_j}^0\|}, \tag{8}$$

where $\mathbf{G}_{T_i}^0$ represents the initial gradient for task $T_i$ at the start of training (i.e., at iteration $t = 0$). This setup dynamically adjusts the scaling based on the task's convergence speed, ensuring that tasks which converge slowly are given larger step sizes, while faster-converging tasks receive smaller updates. This equalization helps slow-converging tasks catch up, promoting balanced learning.

As a result, the optimization of the solving objectives can be adjusted to a weighted loss minimization process, where the loss weight for problem $T_i$ is $\frac{C}{\|\mathbf{G}_{T_i}\|}$. The optimization is then modified as

$$\min_{\theta_s, \bigcup_i \theta_{T_i}} \sum_i \frac{C}{\|\mathbf{G}_{T_i}\|} \mathcal{L}_{T_i}(\theta_s, \theta_{T_i}), \tag{9}$$

which minimizes the solving objectives across all the problems with additional loss weights to balance the gradient magnitudes of different problems.

**Gradient Direction Homogenization.** To homogenize the gradient directions, we follow Javaloy & Valera (2021) to introduce problem-specific rotation matrices on the hidden representations to rotate the optimization landscape, thereby adjusting the optimization directions for gradient consistency. Here we merely focus on the encoder's gradients, leaving the header and decoder gradients problem-specific to allow flexibility and specialization. For each task $T_i$, we introduce a rotation matrix $\mathbf{R}_{T_i} \in SO(d)$ that aligns the task-specific gradients with a unified direction for the shared encoder. Then optimizing the loss calculated with the rotated latent representation $\mathcal{L}_{T_i}(\mathbf{R}_{T_i}\mathbf{z})$ instead of directly optimizing $\mathcal{L}_{T_i}(\mathbf{z})$ can lead to the rotation of the optimization landscapes (Soltanolkotabi et al., 2018), thereby homogenize the gradients across tasks. Here we denote the rotated latent representation as $\mathbf{r}_{T_i} = \mathbf{R}_{T_i}\mathbf{z}$.

Since the rotation matrices introduce additional parameters that only affect the gradient directions for different tasks, we optimize them by minimizing the conflict between the task-specific gradients of the encoder by aligning them toward a common direction. This can be done by maximizing the cosine similarity of the gradients for each task. For task $T_i$, the objective is to minimize:

$$\mathcal{L}^{\text{rot}} = -\sum_i \langle \mathbf{R}_{T_i}^T \nabla_{\mathbf{r}_{T_i}} \mathcal{L}_{T_i}, \mathbb{E}_j(\mathbf{U}_{T_j}) \rangle, \tag{10}$$

where $\mathbf{U}_{T_j}$ denotes the normalized gradients for problem $T_i$, and $\mathbb{E}_j(\mathbf{U}_{T_j})$ is the target direction that all task gradients should point toward, which we select as the average normalized gradient direction across all tasks.

Furthermore, we consider the relationships between different layers of the encoder. Since the first and last layers of the encoder must interface with the input and output features, it is challenging to maintain complete gradient consistency throughout the optimization trajectory for these layers. In contrast, the middle layers have a higher requirement for gradient consistency. In other words, the consistency constraints on the gradients along the optimization trajectories differ across the layers of the encoder. Therefore, we apply different weights to balance the strength of the gradient consistency constraint for different layers. Specifically, we define the gradient consistency loss for the $k$-th layer as $\mathcal{L}_k^{\text{rot}}$, then the layer-weighted objective for the rotation matrices can be written as:

$$\mathcal{L}_{\text{weighted}}^{\text{rot}} = \sum_k w_k \cdot \mathcal{L}_k^{\text{rot}} \tag{11}$$

where $w_k$ indicates the loss weights for different layers. In this paper, we implement a six-layer encoder, and we set $w_1 = w_6 = 0.3, w_2 = w_5 = 0.5, w_3 = w_4 = 1$. With this objective, in each iteration, we additionally involve updating the rotation matrices to balance the gradient directions:

$$\min_{\bigcup_i \mathbf{R}_{T_i}} \sum_k w_k \cdot \mathcal{L}_k^{\text{rot}}(\mathbf{R}_{T_i}) \tag{12}$$

### 4.4 OVERVIEW

Based on the header-encoder-decoder architecture, when optimizing the objective function, we additionally attach rotation matrices for different tasks to transform the features obtained from the encoder. The transformed features are then passed through a problem-specific decoder to obtain the results, and the model is optimized using Eq. 9. On the other hand, in the same step, we update the parameters of the rotation matrix using Eq. 12, optimizing the rotation matrix to ensure that the gradient directions remain consistent across each problem.

In this paradigm, we train the model on data from several problems simultaneously, aiming for the encoder to learn common solving strategies across these problems. The learned strategies are expected to have the potential to solve other similar problems. Therefore, after training, when we encounter a new problem, we do not need to retrain the entire model from scratch. Instead, we can freeze the encoder, hoping that the abstracted solving strategies will already provide good performance. In this case, we only need to retrain the problem-specific single-layer header and decoder. On the other hand, we can also continue to finetune the encoder on different tasks, allowing it to gain more specialized solving abilities for the tasks, thereby improving its performance.

## 5 EXPERIMENTS

In this section, we conduct a series of experiments to evaluate the performance of our proposed method. Specifically, we analyze the relationship between different combinatorial optimization (CO) tasks through gradient similarity, assess the in-distribution solving performance, and investigate the model's generalization ability across tasks, both in a zero-shot setting and with finetuning. The experiments are carried out on six distinct CO problems, including the Travelling Salesman Problem (TSP), Vehicle Routing Problem (VRP), Split Delivery VRP (SDVRP), Orienteering Problem (OP), Prize Collecting TSP (PCTSP), and Stochastic PCTSP (SPCTSP).

### 5.1 EXPERIMENTAL SETUP

**Datasets.** For our TSP, VRP, PCTSP, and OP problems, the common aspect in training and testing instances is the uniform sampling of NN node coordinates within the unit square $[0, 1]^2$. The differences lie in their additional information and objectives. Unlike TSP, other problems explicitly sample a starting point.While VRPs introduces demand at each node and vehicle capacity constraints, PCTSP assigns rewards and penalties to nodes and OP needs the prizes to nodes and the specific type of the prize. All settings mentioned above is a standard procedure as adopted in Kool et al. (2018); Hottung et al. (2021); Joshi et al. (2019); da Costa et al. (2020); Qiu et al. (2022); Sun & Yang (2023); Li et al. (2023). We experiment on the problem scales of 20, 50 for every COP problem.

**Metrics.** Following Kool et al. (2018); Joshi et al. (2019); Sun & Yang (2023); Li et al. (2023), we adopt two evaluation metrics: 1) Obj: the value of the objective function, which is the actual result of the optimization. In a minimization problem, the smaller the objection value, the better the solution, while in a maximization problem, a larger objection value is preferred. 2) Gap: measure the closeness of the solution to the optimal solution, representing the difference between the current solution and the known optimal solution or the theoretical bound.

### 5.2 PROBLEM RELATIONSHIP ANALYSIS AND GENERALIZATION MEASURE

In this experiment, we aim to analyze the underlying relationships between different CO problems by examining the similarity of their gradients during optimization, which can also serve as an indicator of how generalized the learned model is. We train the model simultaneously on classic routing problems, including TSP variants and VRP variants, using the rotation matrices to enforce consistency in the

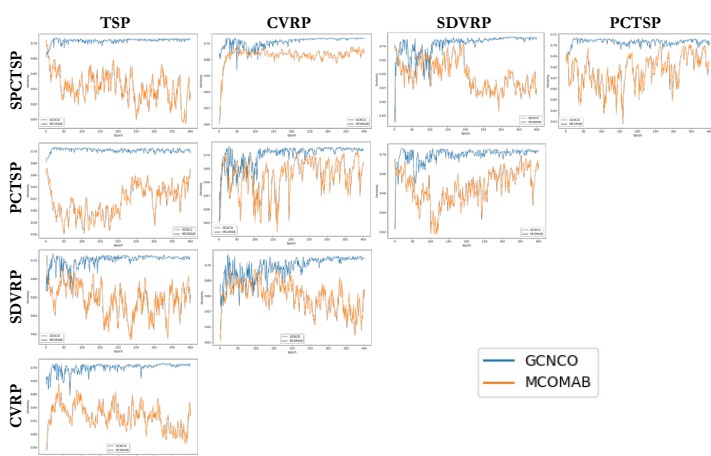

Figure 2: The cosine similarities of the encoder gradients in the training process across problem pairs. The blue curves represent our method, while the orange curves represent the baseline method.

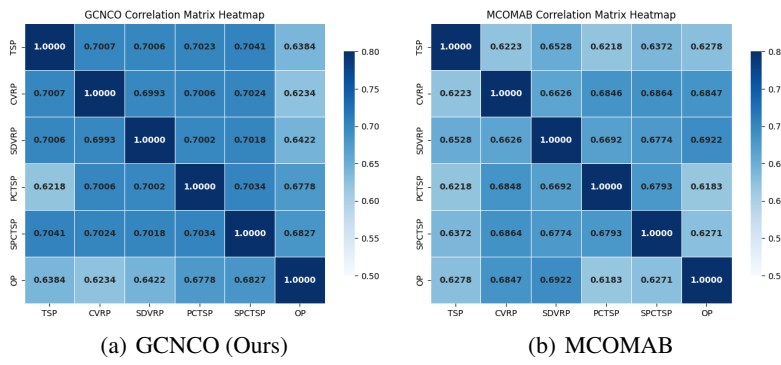

(a) GCNCO (Ours)  (b) MCOMAB

Figure 3: The correlation heatmap involving average cosine similarities of the encoder gradients across problem pairs.

gradient directions across problems. By monitoring the gradients, we observe the degree of alignment between the problems, which reveals the extent of shared structure in the optimization process. The gradient similarity is computed using the cosine similarity of the gradients across tasks, demonstrating how related the optimization landscapes of different CO problems are.

**Results.** Here we primarily compare the baseline Wang & Yu (2023), which also learns across different problems. Fig. ???2 shows the trend of task correlation reflected by the cosine similarities of the encoder gradients across problem pairs. It illustrates the training process in fine detail, where our method not only outperforms the baseline in all task pairs but also converges in the later stages of training, demonstrating its potential to learn inter-task correlations. Fig. ???3 presents the comparison of the problem correlation heatmaps between our method and the baseline method. The problem similarities learned from different models are measured by the average cosine similarities of the encoder gradients across the training process. GCNCO shows improvement across major tasks, which enables learning more inter-task correlations during the training process. On the other hand, we discover a higher correlation between the TSP and VRP variants, while it is more challenging to homogenize OP and other routing problems.

## 5.3 IN-DISTRIBUTION SOLVING PERFORMANCE

In this experiment, we evaluate the model's in-distribution solving performance on multiple CO tasks. We jointly train the model on five out of the six CO tasks and measure its performance on these tasks. The encoder is designed to capture shared strategies, while problem-specific rotation matrices and decoders handle individual task outputs. After the joint training, we then perform additional finetuning on each task individually to assess whether further improvements can be obtained. The performance

Table 1: In-distribution solving performance evaluation with models trained on all six problems and evaluated on individual problems. * indicates the baseline for computing the performance gap.

| Problem | Algorithm | n = 20 | | n = 50 | |
|---|---|---|---|---|---|
| | | Obj. | Gap | Obj. | Gap |
| TSP | LKH3 (Helsgaun, 2017) | 3.84* | 0.00% | 5.70* | 0.00% |
| | MCOMAB-finetuned (Wang & Yu, 2023) | 3.84 | 0.04% | 5.70 | 0.30% |
| | GCNCO-finetuned (Ours) | 3.84 | **0.03%** | 5.70 | **0.27%** |
| CVRP | LKH3 (Helsgaun, 2017) | 6.13* | 0.00% | 10.38* | 0.00% |
| | MCOMAB-finetuned (Wang & Yu, 2023) | 6.18 | 0.86% | 10.59 | 2.15% |
| | GCNCO-finetuned (Ours) | 6.18 | **0.80%** | 10.58 | **2.11%** |
| SDVRP | MCOMAB-finetuned (Wang & Yu, 2023) | 5.96* | **0.00%** | 10.81 | 0.63% |
| | GCNCO-finetuned (Ours) | 6.04 | 1.39% | 10.81* | **0.00%** |
| PCTSP | Gurobi (10s) (Gurobi Optimization, 2020) | 3.13* | 0.00% | 4.48* | 0.00% |
| | MCOMAB-finetuned (Wang & Yu, 2023) | 3.34 | 6.79% | 4.56 | 1.83% |
| | GCNCO-finetuned (Ours) | 3.26 | **4.25%** | 4.54 | **1.64%** |
| SPCTSP | MCOMAB-finetuned (Wang & Yu, 2023) | 3.25 | 0.17% | 4.33* | **0.00%** |
| | GCNCO-finetuned (Ours) | 3.25* | **0.00%** | 4.35 | 0.46% |
| OP | Compass (Kobeaga et al., 2018) | 5.37* | 0.00% | 16.17* | 0.00% |
| | MCOMAB-finetuned (Wang & Yu, 2023) | 5.34 | 0.70% | 16.08 | 0.58% |
| | GCNCO-finetuned (Ours) | 5.34 | **0.46%** | 16.09 | **0.46%** |

is evaluated based on standard metrics such as solution quality and computational efficiency, and we compare the results before and after finetuning to determine the benefits of task-specific adjustments.

**Results.** We compare the baseline model from Wang & Yu (2023) under the same setting, which is also trained on six tasks simultaneously, and is then further tuned on a single task. Additionally, we compare the model's performance with corresponding mainstream heuristic solvers Helsgaun (2017); Gurobi Optimization (2020); Kobeaga et al. (2018). This experiment evaluates the model's in-distribution solving performance, reflecting how well it handles tasks it has been trained on. Since learning generalized strategies may reduce the model's focus on task-specific structures, we further fine-tune the pre-trained models on individual tasks Table 1 illustrates the superiority of GCNCO in solving these tasks. In most cases, GCNCO, when used as a pre-trained model, achieves a smaller gap value or reaches the optimal objective in single-task settings. This demonstrates the models capacity to leverage knowledge from previously encountered tasks, enhancing its performance on individual problems.

## 5.4 Cross-problem Generalization

One of the key strengths of GCNCO lies in its ability to generalize across different CO problems. To evaluate this, we conduct experiments that assess the model's cross-task generalization under two settings: task-specific finetuning and zero-shot generalization with the pretrained encoder.

### 5.4.1 Zero-shot Cross-problem Generalization

In this experiment, we investigate the model's ability to generalize to new tasks without additional training. We freeze the encoder after training it on five CO tasks and directly apply it to solve the sixth, previously unseen task. This zero-shot setting allows us to assess the extent to which the learned shared solving strategies can be transferred to new tasks. We evaluate the model's performance on the new task and compare it to the performance of models trained from scratch on the same task, highlighting the effectiveness of the shared encoder in transferring knowledge across problems.

**Results.** In this experiment, we freeze the parameters of the encoder, which has already been trained on five tasks, and compare its performance with a model trained from scratch using the AM (Kool et al., 2018) framework and the baseline model from (Wang & Yu, 2023), trained under the same conditions. The metrics are evaluated on the left unseen problem for generalization comparison. As shown in Table 2, the objective values and gaps achieved by the frozen encoder on unseen tasks are superior, demonstrating that the multi-task training allowed the model to capture commonalities among different CO tasks. Additionally, when compared to models further fine-tuned on individual problems, we observe that in some cases, additional tuning of the encoder does not yield further performance gains and can already outperform baseline models, including those trained from scratch on specific tasks. This suggests that the learned encoder can obtain plausible generalization performance to new problems.

Table 2: Cross-problem generalization with models trained on five problems an evaluated on the left unseen problem. * indicates the baseline for computing the performance gap.

| PROBLEM | ALGORITHM | n = 20 | | n = 50 | |
|---|---|---|---|---|---|
| | | OBJ. | GAP | OBJ. | GAP |
| TSP | LKH3 (Helsgaun, 2017) | 3.84* | 0.00% | 5.70* | 0.00% |
| | AM (Kool et al., 2018) | 3.85 | 0.08% | 5.71 | 0.35% |
| | MCOMAB-finetuned (Wang & Yu, 2023) | 3.84 | 0.05% | 5.71 | 0.35% |
| | GCNCO-finetuned (Ours) | 3.84 | 0.05% | 5.70 | **0.29%** |
| | GCNCO-frozen (Ours) | 3.84 | **0.04%** | 5.71 | 0.33% |
| CVRP | LKH3 (Helsgaun, 2017) | 6.13* | 0.00% | 10.38* | 0.00% |
| | AM (Kool et al., 2018) | 6.19 | 0.93% | 10.61 | 2.20% |
| | MCOMAB-finetuned (Wang & Yu, 2023) | 6.19 | 0.93% | 10.60 | 2.09% |
| | GCNCO-finetuned (Ours) | 6.18 | 0.87% | 10.59 | 2.13% |
| | GCNCO-frozen (Ours) | 6.18 | **0.82%** | 10.58 | **1.96%** |
| SDVRP | AM (Kool et al., 2018) | 6.10 | 2.34% | 10.84 | 0.22% |
| | MCOMAB-finetuned (Wang & Yu, 2023) | 6.06 | 1.67% | 10.82 | 0.09% |
| | GCNCO-finetuned (Ours) | 5.96* | **0.00%** | 10.82* | **0.00%** |
| | GCNCO-frozen (Ours) | 6.12 | 2.68% | 10.82 | 0.06% |
| PCTSP | Gurobi (10s) (Gurobi Optimization, 2020) | 3.13* | 0.00% | 4.48* | 0.00% |
| | AM (Kool et al., 2018) | 3.36 | 7.36% | 4.80 | 7.14% |
| | MCOMAB-finetuned (Wang & Yu, 2023) | 3.33 | 6.51% | 5.34 | 19.29% |
| | GCNCO-finetuned (Ours) | 3.35 | 7.12% | 4.54 | **1.97%** |
| | GCNCO-frozen (Ours) | 3.31 | **5.78%** | 4.64 | 3.57% |
| SPCTSP | AM (Kool et al., 2018) | 3.24 | 0.34% | 4.63 | 3.04% |
| | MCOMAB-finetuned (Wang & Yu, 2023) | 3.25 | 0.61% | 4.68 | 4.14% |
| | GCNCO-finetuned (Ours) | 3.23* | **0.00%** | 4.50* | **0.00%** |
| | GCNCO-frozen (Ours) | 3.28 | 1.23% | 4.59 | 2.12% |
| OP | Compass (Kobeaga et al., 2018) | 5.37* | 0.00% | 16.17* | 0.00% |
| | AM (Kool et al., 2018) | 5.31 | 1.08% | 16.08 | 0.53% |
| | MCOMAB-finetuned (Wang & Yu, 2023) | 5.34 | 0.51% | 16.10 | **0.43%** |
| | GCNCO-finetuned (Ours) | 5.34 | 0.54% | 16.09 | 0.44% |
| | GCNCO-frozen (Ours) | 5.35 | **0.41%** | 16.08 | 0.52% |

### 5.4.2 FINETUNING FOR CROSS-PROBLEM GENERALIZATION

To further explore the generalization ability of the model, we also evaluate the performance of finetuning the pre-trained encoder on the new task. In this setting, after initially freezing the encoder, we perform additional finetuning on the new task-specific data. This allows us to quantify how well the model adapts to new tasks when a small amount of task-specific data is available. The results of this experiment are compared to the zero-shot setting and the baseline models trained from scratch, demonstrating the encoder's flexibility in handling both new and previously seen tasks.

**Results.** We further fine-tune the learned encoder and compare the AM (Kool et al., 2018) model trained from scratch as well as the fine-tuned MCOMAB baseline (Wang & Yu, 2023). Table. 2 shows that, under the same training setup with five CO tasks, our model achieves an improved gap value on unseen tasks after fine-tuning, and its results are numerically closer to the optimal values provided by heuristic solvers. This demonstrates the effectiveness and adaptability of our method in solving unseen tasks with limited training data.

## 6 CONCLUSION AND FUTURE WORK

This paper presents a novel approach to learning shared-solving strategies across different combinatorial optimization (CO) problems by enforcing gradient consistency along the optimization process. Built upon the header-encoder-decoder architecture which explicitly separates problem-specific components (header and decoder) from a generalized problem-agnostic encoder. The encoder is designed to learn and generalize solving strategies across a diverse set of CO problems by maintaining gradient consistency through the use of problem-specific rotation matrices and loss weights. The effectiveness is demonstrated through extensive experiments on six distinct CO tasks, showing that the proposed approach not only performs well on in-distribution tasks but also exhibits strong generalization capabilities in zero-shot and finetuned cross-problem scenarios. Our results underscore the potential of the learned encoder to support the development of foundational models for CO, as it allows for adaptation to new CO problems without requiring retraining from scratch. This opens up avenues for future work, including extending the method to larger data and exploring specific fine-tuning methods.

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
