# OpenReview forum: "Toward Learning Generalized Cross-Problem Solving Strategies for Combinatorial Optimization"
_ICLR.cc/2025/Conference — ICLR 2025 Conference Withdrawn Submission_

### Official Review · Reviewer_BGDV · 2024-11-02

**Soundness:** 2
**Presentation:** 2
**Contribution:** 2
**Rating:** 5
**Confidence:** 3

**Summary:**

The authors make an effort to address an important problem in the field of combinatorial optimization (CO), specifically the challenge of capturing shared solving strategies across different CO tasks. Their approach, which involves a learning strategy based on gradient consistency, aims to train a neural network with strong generalization capabilities over multiple CO tasks.

**Strengths:**

This manuscript proposes a novel approach to learning shared-solving strategies across different combinatorial optimization (CO) problems by enforcing gradient consistency. This work is a meaningful step forward in in-distribution CO tasks and cross-problem CO scenarios.

**Weaknesses:**

1. While the manuscript argues that gradient consistency is crucial and imposes constraints on gradient
   magnitudes and directions during training, it does not thoroughly explain why gradient consistency
   aids the model in enhancing its generalization abilities across multiple CO tasks. Despite the
   supporting experimental results, a more detailed theoretical explanation, or an intuitive justification
   of the impact of gradient consistency on generalization is needed.

2. The manuscript compares its proposed approach with the MCOMAB and LKH3 methods in the
   experimental
   section. However, it would be beneficial for the authors to briefly introduce these
   methods.
   Providing such background information will help readers who may not be familiar with these
   specific
   methods to better understand the context and significance of the comparisons.

3. The presentation of Figure 2 is suboptimal, with some details and axis scales difficult to
   discern.
   Enlarging the figure to clearly display the relevant details and scale information will
   improve its
   readability and ensure that the graphical data effectively supports the paper’s claims.

4. Although the authors reference relevant literature to describe the experimental settings on
   page 7,
   line 363, a brief recapitulation of these settings within the manuscript would be
   appreciated. This
   would provide immediate contextual understanding for the reader and strengthen the clarity of
   the experimental methodology.

5. The definitions of symbols used throughout the paper are not sufficiently clear. For example, on
   page 4, line 180, the symbols $X_i$ and $Y_i$ are not explicitly defined. Similarly, on page 6,
   line 300, the significance of $SO(d)$ is not explained. Clarifying these notations will enhance
   the readability and comprehension
   of the manuscript.

**Questions:**

See Weaknesses

---

### Official Review · Reviewer_8HVG · 2024-11-03

**Soundness:** 2
**Presentation:** 2
**Contribution:** 2
**Rating:** 5
**Confidence:** 3

**Summary:**

In this paper, the authors propose a deep learning architecture that can solve various types of combinatorial optimization problems. The proposed method generalizes problem solving by learning by distinguishing common and problem-specific parts from various types of problems. When the proposed method was verified through experiments, it showed good performance on various CO problems.

**Strengths:**

It can be said that it has novelty in that it proposes a problem-solving architecture that can generalize and solve various problems by learning the common parts of problem-solving and the individual parts that encode and decode the differences between problems.

In this paper, the authors well organized and compared various types of CO problems through experiments. They designed experiments that can confirm cross-problem generalization.

**Weaknesses:**

The authors have proposed a new method well, but some additional verification seems necessary.
1) It seems necessary to logically explain whether the proposed method can train and converge the encoder and decoder for each problem in the right direction and add related experiments. Although the header-encoder-decoder framework can distinguish the common and different parts of each problem well in terms of its structure and generalize the problem-solving method, it does not seem to guarantee that the desired goal is actually achieved through data-driven learning. It would be good if it were supported by theoretical grounds or experimental proof.
2) In eq (3), it is proposed to learn so that the gradient of the loss is the same for all i and j, but it seems difficult to know whether this term alone guarantees that the problem-solving can be generalized accurately. It would be good to add a detailed explanation for this part.
3) There is insufficient explanation as to why the multi-task learning method proposed in the Javaloy & Valera paper is the most suitable for the method proposed in this paper. If you use other general multi-task learning, it seems that additional explanation is needed for the problems. If there is no special reason to use the method proposed in Javaloy & Valera's paper as it is, the novelty of this paper seems to be weakened.
4) In the Results section on Page 8, typos such as Fig. ???2 and Fig. ???3 were found.

**Questions:**

Questions:
1) Please provide additional explanation referring to the above weakness.
2) What does it mean that five out of six CO tasks were jointly trained in 5.3? Why not six?
3) Is there a reason why the results for the well-known AM algorithm were not compared?
4) What were the results before fine-tuning? It seems that a comparison and analysis of the results before and after fine-tuning is needed.
5) In the zero-shot experiment, for the sixth task, which task did you use the encoder/decoder learned from? It looks that the encoder/decoder for sixth task are not trained.
6) Why is the performance deteriorating in most cases of fine tuning in table 2? It seems that the results are generally difficult to understand.

---

### Official Review · Reviewer_AQ89 · 2024-11-04

**Soundness:** 2
**Presentation:** 2
**Contribution:** 2
**Rating:** 3
**Confidence:** 5

**Summary:**

Summary:  This paper explores a general strategy for solving multi-tasks in CO problems using machine learning. Specifically, these strategies are able to capture shared structural knowledge across tasks in CO problems, making it easier to adapt to related tasks while retaining task-specific capabilities. The article first divides the model architecture into three components: a header, an encoder, and a decoder; where the header and decoder handle problem-specific inputs and outputs, while the encoder is designed to learn shared strategies across different problems. The core of the paper lies in introducing gradient consistency constraints during optimization to ensure that gradient directions and magnitudes remain consistent across different tasks, guiding the model towards a universal solution. Overall, the paper's ideas are clear, and the logic is sound, but some sections may require clarification in the narrative.

**Strengths:**

Strength：
1. The paper considers the issue of multi-task generalization in combinatorial optimization, which has a positive guiding significance for the understanding of machine learning in CO problems.
2. Based on reality, the author decomposes multi-tasks into shared features and task-specific features, which is a common approach in machine learning for multi-tasks, and has a certain research basis when transferred to CO problems.
3. The author's explanation of the problem is concise and easy to read, and a horizontal comparison with similar works is made, making the paper relatively complete.

**Weaknesses:**

Weakness：
1. The fourth section of the paper constitutes the main part of the described method; however, the narratives in 4.1, 4.2, and 4.3 lack logical coherence, exhibiting a strong sense of disconnection, making it challenging to aid in a good understanding of the author's contributions as outlined in the preceding text.
2. The algorithm proposed by the author before Section 4.4 is too similar to the one in [1], lacking sufficient originality.
3.The constraint in Equation (3) on the consistency of gradients across different tasks is expressed too strictly, which is challenging to guarantee in practice.

**Questions:**

Question:
1. The derivation of Equations (6)-(9) is overly vague, and it is unclear how the loss weight for the objective function in Equation (9) is obtained.
2. In Section 4.3, the author mentions the application of rotation matrices to adjust the optimization direction to achieve gradient consistency, but there is a lack of explanation on how these rotation matrices are derived and whether they introduce new training parameters.
3. In the specific loss function derived in Section 4.2, it is not explicitly stated whether $\theta$ in the combination of $\theta_s$ and $\theta_t$ is represented, making it unclear which parameters $\theta$ specifically refers to.
4. There are writing errors like "Fig.???" in the Results section on the eighth page of the article.
5. Among the six CO tasks tested, only the MCOMAB-finetuned algorithm appeared in all tasks, while other comparative algorithms only appeared in individual tasks, without any explanation provided in the paper.
6. In the Zero-shot experiment, the use of the finetuned method by the article introduces unfairness in the comparison process. When compared with the MCOMAB-finetuned algorithm, a significant gap is observed in the PCTSP task, while being surpassed in the OP task, with no explanation provided by the article regarding this discrepancy.


Reference:
[1] Adrian Javaloy and Isabel Valera. Rotograd: Gradient homogenization in multitask learning. arXiv preprint arXiv:2103.02631, 2021.

---

### Official Review · Reviewer_qkjc · 2024-11-13

**Soundness:** 3
**Presentation:** 3
**Contribution:** 2
**Rating:** 5
**Confidence:** 4

**Summary:**

This work proposed a multi-task learning framework for learning combinatorial optimization solvers. The authors designed a header-encoder-decoder architecture where the header and decoder are task-specific and are used to process the inputs and outputs. The encoder is shared by all tasks. The authors utilized an existing work (Javaloy & Valera, 2021) to enforce gradient direction and magnitude consistency. The authors evaluated the effectiveness of the proposed method on six distinct types of CO problems and its superiority over MCOMAB, a previous learning-based CO solver.

**Strengths:**

1. Writing is mostly clear.
2. Results are promising compared with MCOMAB.

**Weaknesses:**

1. The major weakness is the novelty. The multi-task learning setup for CO is not new---it was proposed in MCOMAB paper. The multi-task learning method is not new---the authors used previous work (Javaloy & Valera, 2021). The only technical contribution is the header-encoder-decoder architecture, which from my point of view is not strong enough.

2. What is the computation cost of the proposed header-encoder-decoder architecture, compared with MCOBAT?

3. There are some typos in the writing.

**Questions:**

See the weaknesses part.

---

### Note · Authors · 2025-01-12

I have read and agree with the venue's withdrawal policy on behalf of myself and my co-authors.